# Lessons from COVID-19 syndromic surveillance through emergency department activity: a prospective time series study from western Switzerland

Francois-Xavier Ageron  ,[1] Olivier Hugli  ,[1] Fabrice Dami,[1] David Caillet-Bois,[1] Valerie Pittet,[2] Philippe Eckert,[3] Nicolas Beysard  ,[1] Pierre-Nicolas Carron[1]

[1]Emergency Department, Lausanne University Hospital, Lausanne, Switzerland
[2]Centre for Primary Care and Public Health, University of Lausanne, Lausanne, Switzerland
[3]General Directorate, Lausanne University Hospital, Lausanne, Switzerland

**Correspondence to**
Francois-Xavier Ageron;
francois-xavier.ageron@chuv.ch

## ABSTRACT

**Objective** We aimed to assess if emergency department (ED) syndromic surveillance during the first and second waves of the COVID-19 outbreak could have improved our surveillance system.

**Design and settings** We did an observational study using aggregated data from the ED of a university hospital and public health authorities in western Switzerland.

**Participants** All patients admitted to the ED were included.

**Primary outcome measure** The main outcome was intensive care unit (ICU) occupancy. We used time series methods for ED syndromic surveillance (influenza-like syndrome, droplet isolation) and usual indicators from public health authorities (new cases, proportion of positive tests in the population).

**Results** Based on 37 319 ED visits during the COVID-19 outbreak, 1421 ED visits (3.8%) were positive for SARS-CoV-2. Patients with influenza-like syndrome or droplet isolation in the ED showed a similar correlation to ICU occupancy as confirmed cases in the general population, with a time lag of approximately 13 days (0.73, 95% CI 0.64 to 0.80; 0.79, 95% CI 0.71 to 0.86; and 0.76, 95% CI 0.67 to 0.83, respectively). The proportion of positive tests in the population showed the best correlation with ICU occupancy (0.95, 95% CI 0.85 to 0.96).

**Conclusion** ED syndromic surveillance is an effective tool to detect and monitor a COVID-19 outbreak and to predict hospital resource needs. It would have allowed to anticipate ICU occupancy by 13 days, including significant aberration detection at the beginning of the second wave.

## STRENGTHS AND LIMITATIONS OF THIS STUDY

⇒ A major strength of our study is that we performed and compared time series of surveillance data from the emergency department and the usual surveillance system (health regional authority and laboratory surveillance) during the COVID-19 outbreak.

⇒ Emergency department surveillance data were correlated with intensive care unit occupancy during the outbreak.

⇒ Rigorous methods were applied to detect a significant signal of the second wave before intensive care unit saturation, such as the Early Aberration Reporting System using a Shewhart chart.

⇒ A limitation is that despite a good correlation and early detection of a significant signal, our study lacks external validity.

## INTRODUCTION

In early 2020, the WHO declared the COVID-19 outbreak to be a public health emergency of international concern.[1] Europe was particularly badly hit in spring 2020. After a relative lull in the summer, a second wave occurred in Europe in autumn 2020. Switzerland was among the most affected countries during this period with a much higher COVID-19 incidence compared with the first wave and with a 7-day incidence higher than 600 confirmed cases per 100 000.[2–4] Even if the second wave was expected, its beginning, timing and magnitude were not fully anticipated by the public health authorities.

Current public health surveillance systems include laboratory tests, death rates, hospital-based surveillance and sentinel networks in primary care. Usual surveillance reports use the notification rate of confirmed cases and deaths, laboratory tests, hospital and intensive care unit (ICU) admission and occupancy rates. Primary care sentinel surveillance collects syndromic symptoms related to seasonal influenza-like syndrome. Emergency departments (ED) are uniquely positioned at the interface of the community and hospitals and could serve as early warning systems to identify emerging threats and support decisions of public health authorities.[5] However, only one-third of European countries include ED data in their syndromic surveillance.[6] Until now, this has not been the case in Switzerland where ED data sets have not been used to detect or monitor epidemic outbreaks and, more specifically, the COVID-19 outbreak.

From July to early September 2020, most European countries observed an increase in the incidence of COVID-19 in young people <35 years, but without any significant simultaneous increase in hospital and ICU occupancy. These two concurrent numbers potentially contributed to erroneously reassure public health and political authorities. In addition, a retrospective analysis indicated a persistent higher incidence at the end of the summer in Switzerland, particularly in the western region.[7] However, it remains unknown if the monitoring of cases admitted to the ED would have provided early predictive clues on the resurgence of the pandemic. The aim of our study was to assess if ED syndromic surveillance during the first and second waves of the COVID-19 outbreak could have improved health surveillance and provided additional information for the earlier detection of outbreak signals.

## METHODS

### Study design and population

We did an observational study to assess whether ED syndromic surveillance would have improved the management of the first and second waves of the COVID-19 outbreak in a health system in western Switzerland, based on routine data from the Canton of Vaud and Lausanne University Hospital. The first wave occurred in March 2020 and reached a peak at the beginning of April 2020. The first lockdown in Switzerland started on 17 March 2020 and ended on 11 May 2020. The second wave occurred in November 2020. There was no lockdown applied for the second wave, but some federal restrictions were applied from 3 November 2020, such as restaurant, bar, cinema, museum and library closures.

We used aggregated data from the ED of Lausanne University Hospital, one of the five university hospitals in Switzerland, located in the French-speaking region. It serves as a primary care hospital for the Lausanne area with a population of 250 000 inhabitants and as a tertiary hospital for western Switzerland with a population of 1 million inhabitants. The ED triage includes approximatively 65 000 adult patients per year, two-thirds of whom are admitted to the ED, and one-third to the primary care consultation.

We used data from all consecutive visits leading to ED admission from 25 February 2019 to 19 January 2020 (pre-COVID period used as a control period) and from 25 February 2020 (date of the first infection due to SARS-CoV-2 in Switzerland) to 25 February 2021. Patients referred to the primary care consultation after ED triage were excluded. We also considered aggregated data for the entire population of the Canton of Vaud collected by the emergency medical service (EMS) dispatch centre and the public health authorities.

### Data collection

We collected aggregated data from the ED including date and hour of admission, age group in categories, gender, main complaints at the time of admission classified using the Swiss Emergency Triage Scale,[8] deaths in the ED, hospital admissions to the ward or ICU and positive COVID-19 test notification. The Modified Early Warning Score and the National Early Warning Score (NEWS) were calculated from the initial triage vital signs.[9 10] Data were extracted from the ED patient flow management software (Gyroflux, Lausanne University Hospital, Switzerland) including triage vital signs, symptoms, isolation, ED length of stay, COVID-19 test result, discharge diagnosis and destination of ED patients. Only patient flow aggregated data available in real time were collected, without additional data from medical records. In addition, data were collected from the EMS dispatch centre ('Centrale d'appels sanitaires d'urgence 144') of the Canton of Vaud, including daily emergency calls and ambulance dispatch. We also collected daily hospital occupancy for patients with COVID-19 in general wards and the ICU, as well as data from the Vaud health authority surveillance system (notification of new cases) and laboratory surveillance (results of PCR and antigen tests).[11]

### Outcome

We selected daily absolute ICU occupancy as the primary outcome. ICU beds are a scarce resource requiring trained staff and specific medical devices. The prediction and anticipation of critical care resources has been a key issue in the COVID-19 outbreak. We considered the absolute ICU bed occupancy and not the ICU occupancy rate as the total number of ICU beds regularly evolved during the pandemic, according to needs and available resources (ie, an increase from 35 to 76 beds).

### Surveillance indicators

We studied and compared 'usual' and ED-specific surveillance indicators for COVID-19. Usual surveillance indicators were: (1) number of new confirmed cases of COVID-19 in the population (notification to cantonal public health authorities by medical laboratories or a general practitioner based on a PCR or antigen test); and (2) laboratory surveillance with the proportion of positive tests (PCR and antigen) from all tests performed. ED surveillance indicators were: (1) number of confirmed cases of COVID-19 during ED stay (PCR test); (2) number of patients subjected to droplet isolation measures in the ED; (3) syndromic surveillance with influenza-like syndrome in the ED at triage; (4) number of EMS calls; and (5) number of ambulance dispatches.

### Data analysis

We applied time series analyses for ED COVID-19 visits and for syndromic surveillance, including infectious disease, respiratory disease, cardiac symptoms including chest pain, neurological symptoms including acute paralysis, gastrointestinal bleeding, trauma, psychiatric disorders and hyperglycaemia or hypoglycaemia. We plotted the time series of syndromic surveillance data during the COVID-19 period and compared these to the same

period of 2019. We smoothed time series curves based on the moving 7-day average. We compared graphically ED-EMS surveillance and usual surveillance in the general population and explored the relationship between ICU occupancy and ED-EMS surveillance and traditional surveillance indicators by cross-correlation, and plotted correlograms. We tested the correlation between time series using the Breusch-Godfrey test for higher order serial correlation and Durbin's alternative test for serial correlation.[12 13] The time lag in days between surveillance indicators and ICU occupancy was determined by estimating which lag showed the highest correlation on correlograms. We performed a vector autoregression (VAR) model and considered the optimal time lag for the lowest final prediction error and the lowest Akaike information criterion. We performed Granger causality with a linear regression model and a VAR model to determine which indicator was the best to predict ICU occupancy.[14] Quality control charts were then used to detect early aberration in daily data. The Early Aberration Reporting System uses different methods for temporal aberration detection, including the Shewhart chart (P-chart), moving average and variation of the cumulative sum.[15 16] To assess the usefulness of the ED surveillance system, we assessed graphically the moving average for ED influenza-like syndrome aberrations detected by P-chart during the second wave. The P-chart measures the fraction of nonconforming units in a sample. The control limits for the P-chart were estimated based on the CI of the estimated fraction of the event in the time period using the normal approximation. The formula for the upper and lower limits was: $Pr \pm 3\sqrt{\frac{Pr(1-Pr)}{N}}$, where Pr was the estimated fraction in the time period. Detection of aberration occurs when the value is outside the 99.5% CI. We detail the method in the (online supplemental file 1). We did not reported any missing value for syndromic surveillance in the ED (mandatory item in the software). The sample size was fixed during the study period. We estimated that the minimal sample size was 2668 participants to have a 90% chance of detecting, as significant level of 5%, a difference in the correlation coefficient from 0.75 to 0.80. Data were analysed using Stata V.16.0 (StataCorp, College Station, Texas, USA).

### Patient and public involvement statement

Patients were not involved in the research question and in the design of the study.

### RESULTS

We collected 37 319 ED visits from 25 February 2020 to 25 February 2021 (COVID-19 period) and 42 584 ED visits from 25 February 2019 to 19 January 2020 (pre-COVID (control) period). We reported 1421 (3.8%) confirmed cases of COVID-19 during ED stay, 2181 (5.8%) influenza-like syndromes and 4124 (11.1%) ED visits with droplet isolation (table 1). An increase in influenza-like syndromes was observed during the COVID-19 period. The frequency of ICU admission also increased during the COVID-19 period by 30% (OR 1.30, 95% CI 1.15 to 1.47; p<0.001).

We plotted routine surveillance indicators (confirmed cases, laboratory surveillance and ICU occupancy) and emergency surveillance indicators (EMS and ED indicators) (figure 1). The frequency of positive laboratory tests and confirmed cases first immediately increased, followed by ED influenza-like syndrome, ED isolation droplet and confirmed ED COVID-19. All indicators followed exactly the trend in ICU occupancy with a time lag, depending on the indicators. ED influenza-like syndrome and ED droplet isolation showed a higher increase in the first wave than the second wave compared with ED COVID-19-confirmed cases. All surveillance indicators, except the EMS total number of calls, showed a good correlation with ICU occupancy (table 2). Correlograms showed a positive correlation for all indicators during the second wave (online supplemental file 2). The highest correlations between ED-EMS surveillance indicators and ICU occupancy were obtained with time lags of 10–13 days (table 2). A significant aberration was detected as of 8 March 2020 for the first wave and as of 25 October 2020 for the second wave (figure 2). Aberrations were detected more than 3 weeks before the maximum ICU occupancy was reached. A selection of daily P-charts for ED influenza-like syndrome during the second wave is presented in the online supplemental file 3.

Daily ED activity is presented in figure 3. The total number of ED visits decreased during the first and second waves compared with ED activity in the previous year. Hospital admission remained stable, with a slight increase during the second wave. The number of patients who presented an intermediate to high risk of critical care (NEWS ≥5) increased during the first and second waves. Compared with 2019, trauma, cardiology and stroke activity decreased during the first wave and to a lesser extent during the second wave (online supplemental file 4) (. Gastrointestinal bleeding and diabetes were unchanged during both waves (online supplemental file 5). Allergy decreased during the spring lockdown and increased during the summer break. ED length of stay and waiting time decreased during the first wave (online supplemental file 6). During the second wave, ED length of stay decreased on a smaller scale.

### DISCUSSION

Our study shows the potential for ED syndromic surveillance as an effective tool to detect and monitor COVID-19 outbreaks and to predict hospital resource needs. The ED surveillance system correlated with ICU occupancy and would have allowed to anticipate ICU occupancy by 11–13 days. Of note, it would have also enabled significant aberration detection at the beginning of the second wave. In addition, ED surveillance would provide useful information to plan hospital bed needs, including the number

**Table 1** Patient characteristics

| | COVID period 25 February 2020 to 25 February 2021 | | Previous period 25 February 2019 to 24 February 2020 | |
|---|---|---|---|---|
| | n | % (95% CI) | n | % (95% CI) |
| Total ED visits | 37 319 | | 42 584 | |
| ED influenza-like syndrome | 2181 | 5.8 (5.6 to 6.1) | 235 | 0.6 (0.5 to 0.6) |
| ED isolation droplet | 4124 | 11.1 (10.7 to 11.4) | 510 | 1.2 (1.1 to 1.3) |
| ED respiratory syndrome | 3454 | 9.3 (9.0 to 9.6) | 2713 | 6.4 (6.1 to 6.6) |
| ED COVID-19 confirmed | 1421 | 3.8 (3.6 to 4.0) | – | – |
| ED visits | | | | |
| Medicine | 14 558 | 39.0 (38.5 to 39.5) | 15 261 | 35.8 (35.4 to 36.3) |
| Surgery | 3098 | 8.3 (8.0 to 8.6) | 3799 | 8.9 (8.7 to 9.2) |
| Resuscitation room | 2340 | 6.3 (6.0 to 6.5) | 2201 | 5.2 (5.0 to 5.4) |
| Ambulatory care | 18 491 | 49.5 (49.0 to 50.1) | 22 038 | 51.8 (51.3 to 52.2) |
| Age (years) | | | | |
| 15–29 | 6429 | 17.2 (16.6 to 17.6) | 8567 | 20.1 (19.7 to 20.5) |
| 30–44 | 7571 | 20.3 (19.9 to 20.7) | 8929 | 21.0 (20.6 to 21.4) |
| 45–54 | 4574 | 12.3 (11.9 to 12.6) | 5223 | 12.3 (12.0 to 12.6) |
| 55–64 | 4957 | 13.3 (12.9 to 13.6) | 5258 | 12.3 (12.0 to 12.7) |
| 65–74 | 4496 | 12.0 (11.7 to 12.4) | 4826 | 11.3 (11.0 to 11.6) |
| ≥75 | 8987 | 24.1 (23.6 to 24.5) | 9350 | 22.0 (21.6 to 22.4) |
| Gender (female) | 17 171 | 46.0 (45.5 to 46.5) | 19 745 | 46.4 (45.9 to 46.9) |
| Hospitalisation | 15 325 | 41.1 (40.6 to 41.6) | 15 545 | 36.5 (36.0 to 37.0) |
| ICU | 615 | 1.7 (1.5 to 1.8) | 562 | 1.3 (1.2 to 1.4) |
| MEWS ≥5 | 225 | 0.6 (0.5 to 0.7) | 188 | 0.4 (0.4 to 0.5) |
| NEWS ≥5 | 1028 | 2.7 (2.6 to 2.9) | 849 | 2.0 (1.9 to 2.1) |
| Length of stay in ED ≥6 hours | 18 926 | 50.7 (50.2 to 51.2) | 22 581 | 53.0 (52.6 to 53.5) |
| Death in the ED | 63 | 0.2 (0.1 to 0.2) | 59 | 0.1 (0.1 to 0.2) |

ED, emergency department; ICU, intensive care unit; MEWS, Modified Early Warning Score; NEWS, National Early Warning Score.

and severity of patients admitted to the ED, hospital and ICU admissions and hospital resources required for trauma, cardiology and neurology patients.

## Comparison with other studies

Similar to others, we found a decrease in the total number of ED visits during the first wave.[17–20] Many countries implemented a lockdown during the first wave of the outbreak that explained the decrease in ED visits to a large extent.[21] Importantly, our syndromic surveillance results allow to describe with finer granularity the change in ED activity. During lockdown, we observed a decrease in certain diseases associated with exposure to environmental factors, such as allergy or $CO_2$ emission. Jephcote et al also reported a change in air quality during lockdown in the UK.[22] Kuitunen et al showed that the volume of road traffic and ED visits decreased at the same time and we also observed a reduction in the number of minor and major traumatic injuries.[21] The COVID-19 outbreak well illustrated that a change in human activities contributing to pollution has an immediate effect on population health.

We showed that ED surveillance data were sufficiently accurate to detect changes in the epidemiology of the COVID-19 outbreak, based on our current system using syndromic influenza-like presentations and isolation measures for droplet. In the USA, Pulia et al showed that the surveillance of patients placed in respiratory isolation for an acute respiratory infection was useful to identify and monitor trends during the pandemic.[23] In Paris (France), researchers found that ED visits and EMS calls were correlated with ICU admission, as was the proportion of positive PCR tests.[24] Similar to the Paris study, we showed that ED surveillance predicts ICU occupancy with a time lag of 13 days and the proportion of positive laboratory tests with a lag of 15 days.

## Clinical implications

ED visits have constantly risen during the last decade and one-fourth to one-third of the population visit an ED annually.[25] EDs have become an important player in the public health system and an interface between primary care and the hospital. Indeed, the ED represents today almost the only clinical pathway to unscheduled in-hospital care. For

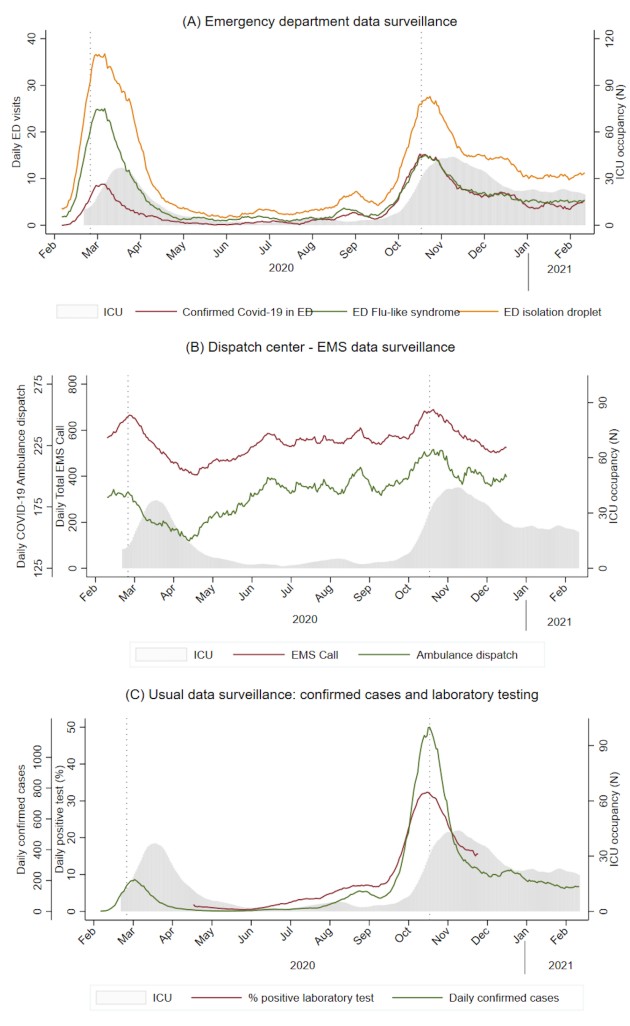

Grey areas represent ICU occupancy at Lausanne University Hospital.

**Figure 1** Time series surveillance indicators. ED, emergency department; EMS, emergency medical service; ICU, intensive care unit.

this reason, the ED has the potential to become a real-time observatory of public health if properly designed with well-defined indicators. Consequently, it is not surprising

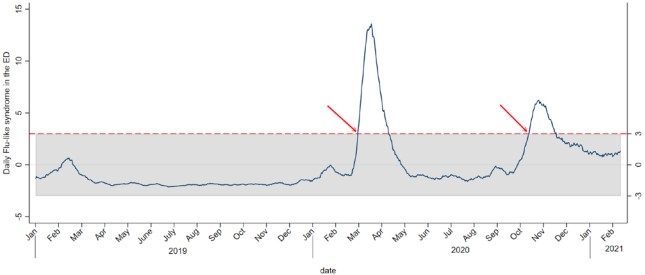

**Figure 2** P-charts of emergency department (ED) influenza-like syndrome during the COVID-19 outbreak at Lausanne University Hospital.

that ED surveillance would have been an effective tool to detect and monitor COVID-19 outbreak activity as it provides simple indicators for real-time monitoring that allow a rapid response from healthcare authorities.

Inside hospitals, ED surveillance would be useful to plan ICU or intermediate care unit resources by predicting ICU occupancy with a significant time lag, specific to the epidemic. Additional syndromic surveillance for surgery and other medical specialities would also be helpful to reduce some activities and reallocate resources where they are the most needed. Of note, ED surveillance would enable to detect the indirect consequences of a pandemic, such as the change in ED visits for life-threatening conditions. It is unlikely that myocardial infarction and strokes decreased during lockdown, but the decrease in chest pain and stroke symptoms observed in the ED suggests that patients avoided attending the ED as a consequence of the 'stay-at-home' campaign and fear of nosocomial COVID-19 infection.[26] This type of ED surveillance data could incite health authorities to inform the population to alert emergency services in case of chest pain and stroke symptoms, regardless of the COVID-19 outbreak.

### Strengths and weaknesses

Our study has some strengths and weaknesses. First, we used a well-described cohort of consecutive ED patient visits without missing data for outcome, syndromic surveillance

| Table 2 | Correlation and time lag between surveillance indicators and ICU occupancy | | | | | |
|---|---|---|---|---|---|---|
| | **Highest correlation coefficient (95% CI)** | **Time lag 1\* (days)** | **Time lag 2\* (days)** | **P value for Breusch-Godfrey test and Durbin's test** | **P value for Granger causality** | |
| | | | | | **Lag 1** | **Lag 2** |
| Confirmed cases | 0.76 (0.67 to 0.83) | 18 | 16 | <0.001 | <0.001 | 0.901 |
| Proportion of positive laboratory tests | 0.92 (0.85 to 0.96) | 15 | 20 | <0.001 | <0.001 | 0.009 |
| EMS call | 0.47 (0.38 to 0.56) | 20 | 7 | <0.001 | <0.001 | 0.368 |
| Ambulance dispatch | 0.33 (0.25 to 0.42) | 33 | 7 | <0.001 | <0.001 | 0.221 |
| ED droplet isolation | 0.79 (0.71 to 0.86) | 11 | 6 | <0.001 | <0.001 | <0.001 |
| ED influenza-like syndrome | 0.73 (0.64 to 0.80) | 13 | 7 | <0.001 | <0.001 | <0.001 |
| ED COVID-19 confirmed | 0.81 (0.73 to 0.88) | 13 | 7 | <0.001 | <0.001 | 0.020 |

*Lag 1 estimated by the highest correlation coefficient on correlograms and lag 2 estimated by the lowest final prediction error (FPE) and the lowest Akaike information criterion (AIC).
ED, emergency department; EMS, emergency medical service; ICU, intensive care unit.

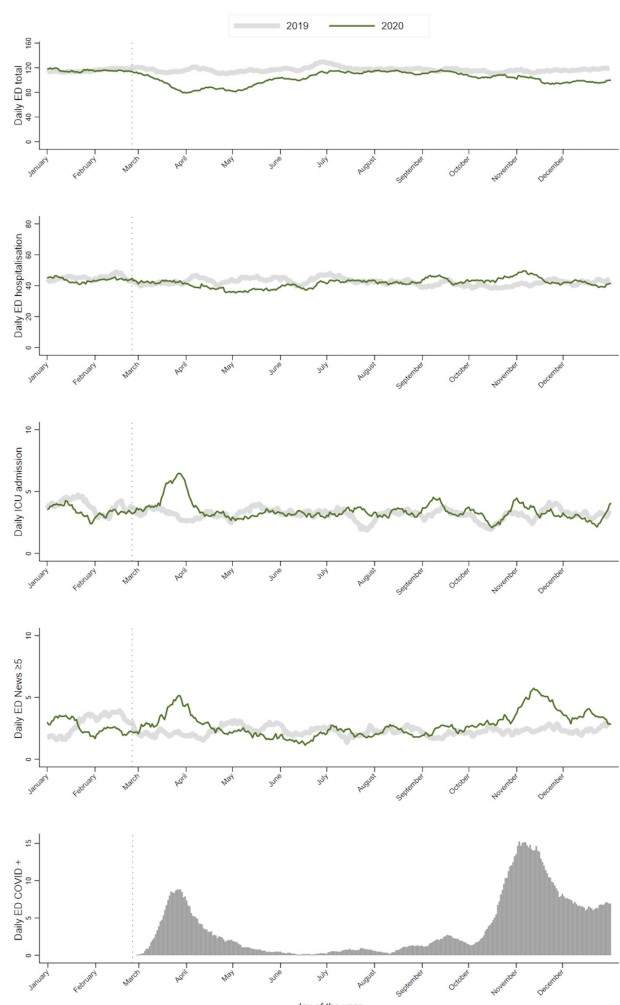

**Figure 3** Time series daily emergency department (ED) general activity at Lausanne University Hospital. ICU, intensive care unit; NEWS, National Early Warning Score.

and triage severity. Follow-up was complete. Second, we used simple observations to assess the obvious correlation between ICU occupancy and surveillance indicators in the first and second waves of the COVID-19 outbreak. We confirmed these observations with rigorous methods used in econometric science and in studies on surveillance systems. Third, we excluded patients attending at ED triage and requiring primary care that might lead to selection bias. However, the objective of the study was to use 'real-life data' available in 'real time' to predict ICU occupancy and detect aberration in syndromic surveillance. Fourth, even if P-charts and the correlation between ICU occupancy and ED surveillance data are obvious in retrospect, surveillance data could be difficult to interpret at the initial stage of a new pandemic and studies to assess the ability of ED surveillance systems to detect a potential new threat need to be performed prospectively in real time. Fifth, our study lacks external validity. The results are dependent on the healthcare system, hospital resources and the triage criteria used in our ED.

In conclusion, ED syndromic surveillance provides additional effective information not accessible in the usual surveillance system. The real-time availability of data makes ED syndromic surveillance a powerful tool for healthcare and political authorities. Future studies on the potential role of emergency services as a public health observatory are needed to further demonstrate their ability to detect and provide data on a larger scale, such as at national level or in situations of infectious diseases, but also in non-infectious diseases related to toxicological, meteorological or psychological diseases.

**Acknowledgements** The authors thank Rosemary Sudan for editorial assistance.

**Author Contributions** F-XA and P-NC designed the study. DC-B and F-XA were responsible for the data management plan. DC-B, FD and VP extracted the data. F-XA was responsible for data analysis. OH, P-NC and F-XA interpreted the data and drafted the manuscript. NB, VP, DC-B, FD, PE, OH, P-NC and F-XA contributed to the interpretation of the results and critical revision of the manuscript, and approved the final version. NB, VP, DC-B, FD, PE, OH, P-NC and F-XA agreed to be accountable for all aspects of the work. F-XA is responsible for the overall content as the guarantor and accepts full responsability.

**Funding** The authors have not declared a specific grant for this research from any funding agency in the public, commercial or not-for-profit sectors.

**Competing interests** None declared.

**Patient and public involvement** Patients and/or the public were not involved in the design, or conduct, or reporting, or dissemination plans of this research.

**Patient consent for publication** Not required.

**Ethics approval** This study was assessed by the Ethics Review Board of the Canton of Vaud (reference number: CER-VD 2020-00731) that decided to waive the need for approval and need for informed consent as this study collected only aggregate data and no individual patient-level data.

**Provenance and peer review** Not commissioned; externally peer reviewed.

**Data availability statement** Data are available upon reasonable request. Data are available on reasonable request and with agreement from Lausanne University Hospital and Public Health Authorities of the Canton of Vaud.

**ORCID iDs**
Francois-Xavier Ageron http://orcid.org/0000-0003-0520-3619
Olivier Hugli http://orcid.org/0000-0003-2312-1625
Nicolas Beysard http://orcid.org/0000-0003-3561-1250

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
