## [Reviewer comments · BMJ Open]

ARTICLE DETAILS

TITLE (PROVISIONAL)	Lessons from COVID-19 syndromic surveillance through emergency department activity: a prospective time-series study from western Switzerland
AUTHORS	AGERON, Francois-Xavier; Hugli, Olivier; Dami, Fabrice; Caillet-Bois, David; Pittet, Valerie; Eckert, Philippe; Beysard, Nicolas; Carron, Pierre-Nicolas

VERSION 1 – REVIEW

REVIEWER	GHAZALI, Daniel Aiham Hospital Bichat - Claude-Bernard, Emergency Department
REVIEW RETURNED	20-Jul-2021

GENERAL COMMENTS	Dear editor in chief, dear authors Thank you for the opportunity to review the article entitled: 'Lessons from COVID-19 syndromic surveillance through emergency department activity: a prospective time-series study from western Switzerland'. The authors aimed to assess if emergency department (ED) syndromic surveillance could be used as an effective tool to detect and monitor a COVID-19 outbreak and to predict health surveillance. However, the impact of this study is limited by several factors, including the monocentric design. Revisions are required. Major points Title Suggest rewriting the title. What makes the study prospective? It is a monocentric study Methods Page 5, Line 48: Why were patients, referred to the primary care consultation after ED triage, were excluded? I speculate that patients with influenza-like syndrome without severe criteria are seen at the fast track level and therefore in this area. These data could be essential in the analysis and anticipation of the evolution of the pandemic. This exclusion must be justified by which method was confirmed the cases of covid? Is it by a biological test? is it at the ED (point-of-care) or laboratory level? Page 8, Line 3: - By which method was confirmed the cases of covid? Is it by a biological test? is it at the ED (point-of-care) or laboratory level? - Could authors comment on access to COVID-19 testing among ED during the study period as it would have impacted the reported COVID-cases? Were there any racial and socio-economic disparities (poor individuals, homeless etc.) in access to testing?
---

	These are critical to understand whether these played a role in visits to ED or to seek medical care. Page 8, Line 3: data analysis The main research method of the article is to use interrupted time series analysis to evaluate the changes in the number of EDs and syndromic surveillance. However, interrupted time series (ITS) design is mostly used for evaluating the effectiveness of intervention measures. Is the interrupted time series the best statistical method to use in this type of study? Can a statistician justify the relevance of this test in this study since other tests exist for this type of analysis? Results Page 3, Line 3: it is not certain that the results are the same between the first and the second waves since patient management and biological diagnosis have evolved over time I suggest doing an analysis for each wave separately and testing this hypothesis Page 9, Line 27: The authors should distinguish between lockdown periods and compare activity during the pandemic and outside the lockdown period, and the period between the two waves. It is very likely that ED activity is different during lockdown. Page 9, Line 36: the size of the population must be specified Discussion Page 9, Line 45: Is the first wave similar to the second one? Not sure. The definition has changed as the pandemic as evolved. Methods and time required to make the diagnosis changed between the first and second waves. This factor may influence the prediction results and may be different between the waves Page 9, line 47: how to interpret an aberration? This point must be developed. Page 9, Line 50: this point should be developed in the discussion What was the difference between the two waves? Why only during the second wave Page 10, Line 42: This point should be developed: why it is important to predict ICU occupancy with time lag of two weeks? Page 12, Line 3: this is a major limitation. The author should analysis data of other ED in the region as has been done in the literature Minor points Introduction Page 6, Line 8: authors should indicate the start and end dates of the first and second waves of covid and specify the start and end dates of lockdown Methods Page 6, Line 23: I do not understand the use of the data of the regulation center of Vaud. Are the calls from Lausanne managed by this center? Calls from this center requiring an ambulance transport are dispatched to which hospitals? The authors should clarify this information Page 8, Line 8: where are these ambulances dispatched? Results Page 3, Line 5: what about the first wave? (Figure 2 present only results of the second wave) Discussion Page 10, Line 14: Suggest beginning the discussion with major results and then to continue with this section Tables and Figures Figure 1: needs to be re-edited to be more readable
--	--

	Figure 2: this figure is unreadable and difficult to understand. Perhaps we should focus on the aberrations Figure 3: it would be better to choose the periods of the study (i.e., 25 Feb, 2019 to 19 Jan, 2020 and 25 Feb, 2020 to 25 Feb, 2021) Once again, thank you for the opportunity to review this article. Best regards
--	---

REVIEWER	Leibner, Evan Icahn School of Medicine at Mount Sinai, Emergency Medicine and Institute for Critical Care
REVIEW RETURNED	01-Aug-2021

GENERAL COMMENTS	Overall an interesting idea and novel findings that would be helpful in helping to plan for ICU beds. The statistical methods used in this paper should have a statistical review.
--

REVIEWER	Costa-Santos, Cristina Faculty of Medicine, University of Porto, Centre for Research in Health Technologies and Information Systems (CINTESIS), Information Sciences and Decision on Health Department (CIDES)
REVIEW RETURNED	11-Nov-2021

GENERAL COMMENTS	The authors assessed whether emergency department syndromic surveillance during the first and the second waves of the COVID-19 outbreak could have improved the surveillance system and provided additional information for the earlier detection of outbreak signals. It is an important study, however the authors should discuss how the large sample sizes, like the one used in their study, can transform small effects in statistically significant effects. The study does not propose any method of predicting the occupation of the UCI in real time. The association found was already somewhat expected, so the authors could propose the methodology to be used in future studies to assess the ability of ED surveillance systems to detect potential new threat.
--

VERSION 1 – AUTHOR RESPONSE

Reviewer #1

We thank the reviewer for the constructive comments provided.

Dr. Daniel Aiham GHAZALI, Hospital Bichat - Claude-Bernard, Université Paris 7 Diderot Comments to the Author:

Dear editor in chief, dear authors

Thank you for the opportunity to review the article entitled:

'Lessons from COVID-19 syndromic surveillance through emergency department activity: a prospective time-series study from western Switzerland'.

The authors aimed to assess if emergency department (ED) syndromic surveillance could be used as an effective tool to detect and monitor a COVID-19 outbreak and to predict health

surveillance. However, the impact of this study is limited by several factors, including the monocentric design. Revisions are required.

Major points

Title

Suggest rewriting the title. What makes the study prospective? It is a monocentric study Methods

Response: The first protocol of the study submitted to the regional ethics committee was dated 23 March 2020 and was accepted by the hospital commission on 8 April 2020. We collected data prospectively for this study and checked completeness and accuracy of the data every week. The statistical analysis plan was written in May 2020.

Page 5, Line 48: Why were patients, referred to the primary care consultation after ED triage, were excluded? I speculate that patients with influenza-like syndrome without severe criteria are seen at the fast track level and therefore in this area. These data could be essential in the analysis and anticipation of the evolution of the pandemic. This exclusion must be justified by which method was confirmed the cases of covid? Is it by a biological test? is it at the ED (point-of-care) or laboratory level?

Response: Patients referred to the primary care consultation were excluded because they were not admitted to the emergency department (ED). Thus, patients were not recorded in the ED database. Patients classified as non-urgent (class 4 or 5 of the ED triage scale) were not admitted to the ED and referred to one of the many ambulatory care services available in Switzerland. It is true that this study only includes patients presenting an influenza-like syndrome with a severe or moderate criterion. The aim of the study was not to substitute population surveillance by ED surveillance, but to assess if ED data surveillance provides additional information, including early prediction of ICU occupancy. An increased incidence in the population is not necessary associated with an increase of hospital or ICU occupancy. The ED represents the interface between the community and hospitals. Omitting patients with less severity is not a limitation as they are not admitted to hospital and do not impact on hospital resources.

Page 8, line 3:

- By which method was confirmed the cases of covid? Is it by a biological test? is it at the ED (point-of-care) or laboratory level?

Response: COVID cases in the ED were confirmed by a PCR laboratory test. COVID cases in the population of the canton of Vaud were reported by laboratory test (PCR or antigen test) to the regional health authorities. We have now added this detail in the text:

“(notification to cantonal public health authorities by medical laboratories or a general practitioner based on a PCR or antigen test)”

and

“(1) number of confirmed cases of COVID-19 during ED stay (PCR test)”.

- Could authors comment on access to COVID-19 testing among ED during the study period as it would have impacted the reported COVID-cases? Were there any racial and socio-economic disparities (poor individuals, homeless etc.) in access to testing? These are critical to understand whether these played a role in visits to ED or to seek medical care.

Response: It is clear that COVID-19 testing was at first difficult. For this reason, we used a flu-like syndrome (clinical syndrome) that is not impacted by laboratory tests. Figure 1 shows the discrepancy between a flu-like syndrome and confirmed COVID-19 during the first wave. In the second wave, confirmed COVID-19 and a flu-like syndrome were similar.

We did not collect racial and socioeconomic data for routine surveillance. However, it is unlikely that social or racial inequalities affected testing in the ED. Medical insurance is mandatory for all people living in Switzerland and laboratory tests were free for all persons during the study period. Individuals without insurance (mostly migrants) benefit from free care only in the ED. A flu-like syndrome is not impacted by testing access or hospital admission.

Page 8, Line 3: data analysis

The main research method of the article is to use interrupted time series analysis to evaluate the changes in the number of EDs and syndromic surveillance. However, interrupted time series (ITS) design is mostly used for evaluating the effectiveness of intervention measures.

Is the interrupted time series the best statistical method to use in this type of study?

Can a statistician justify the relevance of this test in this study since other tests exist for this type of analysis?

Response: The main research method of our study was not to use interrupted time series analysis and we kindly suggest that the reviewer has made a confusion in the reading of the manuscript.

On page 8, we wrote: "We applied time series analyses for ED COVID-19 visits and for syndromic surveillance". Interrupted time series is different from time series analysis with quality control charts, such as the Shewhart chart. These analyses represent the best statistical method for surveillance data.

Results

Page 3, Line 3: it is not certain that the results are the same between the first and the second waves since patient management and biological diagnosis have evolved over time I suggest doing an analysis for each wave separately and testing this hypothesis

Response: Figure 1 clearly demonstrated that that ED surveillance (including flu-like syndrome that is not impacted by biological diagnosis) followed ICU occupancy and regional health authority surveillance very well. Given the obvious relationship between ED surveillance and ICU occupancy, separate analyses are unlikely to provide useful information. In the case of a weak correlation between ED surveillance and ICU occupancy, separate analyses would have been interesting to identify bias toward the null due to a weak correlation in a specific wave.

Page 9, Line 27: The authors should distinguish between lockdown periods and compare activity during the pandemic and outside the lockdown period, and the period between the two waves. It is very likely that ED activity is different during lockdown.

Response: Time series analysis is an analysis of data collection over a period of time. We stated in Methods that we collected data from 25 February, 2020, to 25 February, 2021 (with a control period in 2019). This period includes several waves of COVID-19, different lockdowns and inter-lockdown periods. Figures 3 and 4 compare ED activity during and after the first lockdown. ED activity decreased during lockdown and increased after and then reached the same level as in 2019. On p. 9,

we wrote: "Daily ED activity is presented in figure 3. The total number of ED visits decreased during the first and second waves compared to ED activity in the previous year."

Page 9, Line 36: the size of the population must be specified

Response: We are not sure that we have understood the information required by the reviewer. The different figures (i.e., 3, 4 and supplement) present the number of daily patient admissions. We have reported the size of the population admitted to the ED by time period at the beginning of the Results section. We also reported the size of the global population in the Methods section.

Discussion Page 9, Line 45: Is the first wave similar to the second one?

Not sure. The definition has changed as the pandemic as evolved. Methods and time required to make the diagnosis changed between the first and second waves. This factor may influence the prediction results and may be different between the waves

Response: The definition of a flu-like syndrome did not change during the study period. The confirmed cases of COVID-19 did not change as they corresponded to a PCR or antigen laboratory test. It is true that the time to obtain confirmed laboratory cases was shorter in the second wave, which explains why ED confirmed cases were similar to flu-like syndrome cases in the second wave (figure 1). However, cross-correlation and serial correlations were not affected as correlation was higher for ED confirmed cases than flu-like syndrome cases.

Page 9, line 47: how to interpret an aberration? This point must be developed.

Response: This point was developed in detail in the Methods section. However, we thank the reviewer for highlighting the aberration definition as we inadvertently included a wrong formula in the Methods section and for which we sincerely apologise. Indeed, we omitted the number 3 before the interval of prediction:

"To assess the usefulness of the ED surveillance system, we assessed graphically the moving average for ED flu-like syndrome aberrations detected by P-chart during the second wave. The P-chart measures the fraction of nonconforming units in a sample. The control limits for the P-chart were

estimated using the formula: $Pr \pm 3$ where Pr is the estimated fraction."

As aberration detection is perhaps difficult to understand for the general reader, we have now added the following text:

"The control limits for the P-chart were estimated based on the confidence interval of the estimated fraction of the event in the time period using the normal approximation. The formula for the upper and

lower limit was : $Pr \pm$ where Pr was the estimated fraction in the time period. Detection of aberration occurs when the value is outside the 99.5% confidence interval."

Page 9, Line 50: this point should be developed in the discussion What was the difference between the two waves? Why only during the second wave

Response: There was no difference between the two waves in terms of correlation between ED surveillance and ICU occupancy as we used time series (and not interrupted time series). Our study is a prospective study. For the first wave, the detection of aberration was not implemented. We implemented the surveillance system in April 2020, thus allowing to detect aberration only for the next wave.

Page 10, Line 42: This point should be developed: why it is important to predict ICU occupancy with time lag of two weeks?

Response: This point was stated in the paragraph regarding the comparison with other studies. We developed this point in the clinical implications paragraph: "Inside hospitals, ED surveillance would be useful to plan ICU or intermediate-care unit resources by predicting ICU occupancy with a significant time lag, specific to the epidemic."

Page 12, Line 3: this is a major limitation. The author should analysis data of other ED in the region as has been done in the literature

Response: We agree with the reviewer. This is a limitation that we stated clearly in the manuscript. However, it is important to not confound generalizability and representativeness. Our study is not representative for other systems of care in Europe or worldwide. By contrast, the fact that the ED surveillance system is useful to detect ICU saturation in the next 2 weeks is generalizable as the ED is the first step of the patient care pathway to the ICU in many healthcare systems.

Minor points

Introduction

Page 6, Line 8: authors should indicate the start and end dates of the first and second waves of covid and specify the start and end dates of lockdown

Response: Thank you for this comment. We have now added the following text:

"The first wave occurred in March 2020 and reached a peak at the beginning of April 2020. The first lockdown in Switzerland started on 17 March 2020 and ended on 11 May 2020. The second wave occurred in November 2020. There was no lockdown applied for the second wave, but some federal restrictions were applied from 3 November 2020, such as restaurant, bar, cinema, museum and library closures."

Methods Page 6, Line 23: I do not understand the use of the data of the regulation center of Vaud. Are the calls from Lausanne managed by this center? Calls from this center requiring an ambulance transport are dispatched to which hospitals? The authors should clarify this information Page 8, Line 8: where are these ambulances dispatched?

Response: Prehospital emergency data surveillance might be useful. Indeed, if ambulance dispatch or emergency calls increase, it could be associated with a specific event such as the COVID-19 pandemic. The results of this study showed that dispatch centre data were poorly correlated to the COVID outbreak. The emergency dispatch centre covered the canton of Vaud. Ambulances are dispatched from the different areas of the canton and transport patients to different hospitals. Lausanne is the tertiary hospital of the area. However, it does not matter where these ambulances are dispatched or where patients are transported. This study assesses if some emergency routine data are useful to outbreak detection. Switzerland is a federal state and each state (canton) has an emergency dispatch centre and several hospitals. For the reviewer, it is similar to the French system where there is one "SAMU" by administrative department and several hospitals in that department.

Results

Page 3, Line 5: what about the first wave? (Figure 2 present only results of the second wave)

Response: This point was already answered above. For the first wave, the detection of aberration was not implemented. We implemented the surveillance system in April 2020, thus allowing to detect aberration only for the second wave. However, following the reviewer's request for a modification of figure 2 (below), we have now added the detection of aberration at the first wave in the new figure 2 and modified the text as follows:

"A significant aberration was detected as of 8 March 2020 for the first wave and as of 25 October 2020 for the second wave (figure 2). Aberrations were detected more than three weeks before the maximum ICU occupancy was reached. A selection of daily P-charts for ED flu-like syndrome during the second wave are presented in the supplement (efigure 5)."

Discussion Page 10, Line 14: Suggest beginning the discussion with major results and then to continue with this section

Response: The Discussion section started with the main results and then continued with comparisons with other studies.

Tables and Figures

Figure 1: needs to be re-edited to be more readable

Response: Thank you for this comment. We have now re-edited the figure and simplified the axis.

Figure 2: this figure is unreadable and difficult to understand. Perhaps we should focus on the aberrations

Response: We have now changed figure 2 by using only one global P-chart including data from 2019 to 2021 and showing the P-chart detection limits of the aberrations. However, we believe that it has lost the dynamic of aberration detection in ED daily surveillance. Of note, the two other reviewers did not report any difficulty to read or understand this figure. However, we have added a new figure 2 in the submission system and placed the previous figure in the online-supplement if the editor agrees.

Figure 3: it would be better to choose the periods of the study (i.e., 25 Feb, 2019 to 19 Jan, 2020 and 25 Feb, 2020 to 25 Feb, 2021)

Response: We chose the same periods in 2019 and 2020 that appeared to be quite similar.

Manuscript # bmjopen-2021-054504 entitled "Lessons from COVID-19 syndromic surveillance through emergency department activity: a prospective time-series study from western Switzerland"

Reviewer #2

We thank the reviewer for the appreciation of the interest of our work.

Dr. Evan Leibner, Icahn School of Medicine at Mount Sinai
Comments to the Author:

Overall an interesting idea and novel findings that would be helpful in helping to plan for ICU beds. The statistical methods used in this paper should have a statistical review.

Manuscript # bmjopen-2021-054504 entitled "Lessons from COVID-19 syndromic surveillance through emergency department activity: a prospective time-series study from western Switzerland"

Reviewer #3

We thank the reviewer for the appreciation of the importance of our study and the pertinent comments provided.

Dr. Cristina Costa-Santos, Faculty of Medicine, University of Porto Comments to the Author:

The authors assessed whether emergency department syndromic surveillance during the first and the second waves of the COVID-19 outbreak could have improved the surveillance system and provided additional information for the earlier detection of outbreak signals.

It is an important study, however the authors should discuss how the large sample sizes, like the one used in their study, can transform small effects in statistically significant effects.

The study does not propose any method of predicting the occupation of the UCI in real time. The association found was already somewhat expected, so the authors could propose the methodology to be used in future studies to assess the ability of ED surveillance systems to detect potential new threat.

Response: The sample size is not large. Annually, it represents 40,000 patients in total. However, daily surveillance of flu-like syndrome represents less than 10 patients daily.

We agree with the reviewer that we did not propose a method of predicting ICU occupancy and we used the usual method for surveillance data, such as the early aberration reporting system (EARS) using the Shewhart chart. In response to the reviewer's remark, we have now added a practical method ready to use in the online supplement file.

eSupplement 6. Method for Early Aberration Reporting System (STATA computing)

1. Identify the flu-like syndrome in emergency department software using the main complaint reported by the patient: combine the flu-like syndrome or fever/cough/shortness of breath and isolation droplet depending on the main complaint collected.

```
.gen flu_like_syndrome=1 if fever==1 | shortness_breath==1 | cough==1 & isolation_droplet
```

1. Extract the number of flu-like syndromes by day and the total number of patients attending the ED by day.

```
.collapse (sum) flu_like_syndrome total_ed id, by(date_admission)
```

1. Perform a moving average for the number of flu-like syndromes by day (7 days or less).

```
.tssmooth ma flu_like_sd_ma= flu_like_syndrome, window(7 1 7)
```

```
.tssmooth ma total_ed_ma= total_ed, window(7 1 7)
```

1. Plot the P-chart (using the usual limit $Pr \pm$).

```
. pchart flu_like_sd_ma date_admission total_ed_ma, sta recast(line) ytitle("Daily Flu-like syndrome in the ED",size(small)) title("P-chart moving average fraction defective", size(small)) xtitle("date",size(small)) ylabel(, labsize(small) axis(1)) ylabel(, labsize(small))
```

We have also modified the strengths and limitations section following the abstract in order to focus on the practical methodology used.

- We performed and compared time series of surveillance data from Emergency Department and usual surveillance system (health regional authority and laboratory surveillance) during the Covid-19 outbreak.
- We study correlation between Emergency department surveillance data and Intensive Care Unit occupancy during Covid-19 outbreak.
- We used rigorous method such as Early Aberration Reporting System using Shewhart chart to detect significant signal of the second wave before ICU saturation.
- Despite a good correlation and early detection of significant signal, our study lacks external validity
- Surveillance data could be difficult to interpret at the initial stage of a new pandemic and studies to assess the ability of ED surveillance systems to detect a potential new threat need to be perform in real time.

VERSION 2 – REVIEW

REVIEWER	Costa-Santos, Cristina Faculty of Medicine, University of Porto, Centre for Research in Health Technologies and Information Systems (CINTESIS), Information Sciences and Decision on Health Department (CIDES)
REVIEW RETURNED	31-Dec-2021
GENERAL COMMENTS	Authors improve their manuscript, however, in some analyses performed, the sample size is, in fact, large, which is reflected, for example, in very narrow confidence intervals. A large sample size can give us more power to detect tiny differences. Authors should calculate the appropriate sample size or at least discuss this issue.

VERSION 2 – AUTHOR RESPONSE

Reviewer: 3

Dr. Cristina Costa-Santos, Faculty of Medicine, University of Porto

Comments to the Author:

Authors improve their manuscript, however, in some analyses performed, the sample size is, in fact, large, which is reflected, for example, in very narrow confidence intervals. A large sample size can give us more power to detect tiny differences. Authors should calculate the appropriate sample size or at least discuss this issue.

We add the following in the method section: “ The sample size was fixed during the study period. We estimated that the minimal sample size was 2,668 participants to have a 90% chance of detecting, as significant level of 5%, a difference in the correlation coefficient from 0.75 to 0.80.”